# Photobiomodulation therapy in improvement of harmful neural plasticity in sodium salicylate-induced tinnitus

**Katayoon Montazeri[1], Mohammad Farhadi[1], Abbas Majdabadi[2], Zainab Akbarnejad[1], Reza Fekrazad[3], Ali Shahbazi[4], Saeid Mahmoudian[1] ***

1 The Five Senses Health Institute, ENT and Head and Neck Research Center, School of Medicine, Iran University of Medical Sciences (IUMS), Tehran, Iran, 2 Dentistry Research Institute, Laser Research Center of Dentistry, Tehran University of Medical Sciences, Tehran, Iran, 3 Radiation Sciences Research Center, Laser Research Center in Medical Sciences, AJA University of Medical Sciences, Tehran, Iran, 4 Faculty of Advanced Technologies in Medicine, Department of Neuroscience, Iran University of Medical Sciences (IUMS), Tehran, Iran

* mahmoudian.s@iums.ac.ir, saeid.mahmoudian@gmail.com

**Data Availability Statement:** Data Statement: the contact point for the data access would be the auditory neuroscience research core of ENT and Head and Neck Research Center, Iran University of

## Abstract

Tinnitus is a common annoying symptom without effective and accepted treatment. In this controlled experimental study, photobiomodulation therapy (PBMT), which uses light to modulate and repair target tissue, was used to treat sodium salicylate (SS)-induced tinnitus in a rat animal model. Here, PBMT was performed simultaneously on the peripheral and central regions involved in tinnitus. The results were evaluated using objective tests including gap pre-pulse inhibition of acoustic startle (GPIAS), auditory brainstem response (ABR) and immunohistochemistry (IHC). Harmful neural plasticity induced by tinnitus was detected by doublecortin (DCX) protein expression, a known marker of neural plasticity. PBMT parameters were 808 nm wavelength, 165 mW/cm$^2$ power density, and 99 J/cm$^2$ energy density. In the tinnitus group, the mean gap in noise (GIN) value of GPIAS test was significantly decreased indicated the occurrence of an additional perceived sound like tinnitus and also the mean ABR threshold and brainstem transmission time (BTT) were significantly increased. In addition, a significant increase in DCX expression in the dorsal cochlear nucleus (DCN), dentate gyrus (DG) and the parafloccular lobe (PFL) of cerebellum was observed in the tinnitus group. In PBMT group, a significant increase in the GIN value, a significant decrease in the ABR threshold and BTT, and also significant reduction of DCX expression in the DG were observed. Based on our findings, PBMT has the potential to be used in the management of SS-induced tinnitus.

## Introduction

Tinnitus or the buzz, is the phantom perception of sound that occurs due to a sensory deprivation that leads to secondary central compensatory changes [1] It is estimated that nearly 14% of the world's population has experienced tinnitus, more than 2% of which is the severe form.

Medical Sciences. Because all the intellectual rights of this study belong to the Iran University of Medical Sciences, it cannot be transferred to a third party without the written permission of the university, and the data cannot be shared publicly. Data are available to researchers who meet the confidential data access criteria from the Ethics Committee/Institutional Data Access of auditory neuroscience research core of ENT and Head and Neck Research Center, Iran University of Medical Sciences (contact via Email address: enthns@iums.ac.ir; Tel.: +98 21 66552828, +98 21 66504294; Fax: +98 21 66525329; Website: https://enthns.iums.ac.ir/ Correspond Email addresses: mahmoudian.s@iums.ac.ir; saeid.mahmoudian@gmail.com).

**Funding:** All the authors whose names are included in this article acknowledge that parts of this study were provided with the financial support of two institutes, INSF (reference number 91058320/insf/98020384) and Iran University of Medical Sciences (registration number of 98-2-22-15523), for which they are grateful.

**Competing interests:** The authors have declared that no competing interests exist.

The quality of life of approximately 120 million people worldwide is severely impaired by tinnitus [2]. As a symptom, it can be associated with auditory, metabolic, psychiatric, inflammatory and traumatic underling disorders or using ototoxic drugs [3,4].

Tinnitus can be associated with normal hearing, hearing loss, and hidden hearing loss. In hidden hearing loss, despite the normal hearing level in routine audiological tests, cochlear synaptopathy, which refers to the damage of synapses between inner hair cells and auditory nerve fibers in the cochlea, has been developed [5]. In the auditory brainstem response (ABR) test in animal models, a decrease in the amplitude of wave I as well as a change in summating potential (SP)/ action potential (AP) ratio has been suggested as signs of hidden hearing loss [6–8].

The most important problems of tinnitus are the lack of an objective method to evaluate it and the lack of an accepted and effective treatment for it [9]. In recent years, progress has been made in both the objective assessment and treatment of tinnitus using animal models, but to confirm the results of these animal studies, human controlled studies and high quality RCTs need to be performed [10–12]. In most of the tinnitus subjects, a neurological change starts in the cochlea and spreads to the central nervous system (CNS) [13]. Changes in the topography of the cerebral cortex, imbalance of excitatory/inhibitory processes and disruption of functional brain connectivity, as well as increased spontaneous firing rates and neural synchrony are the common neurological changes which have been reported in tinnitus [14].

Noise cancelling deficit, lack of habituation to repeated non-informative sounds, occurrence of central compensatory changes due to a lack of auditory input, disability of the limbic system to regulate buzzing and a bottom-up deafferentation have been proposed as the mechanism of tinnitus [15–19].

Neural plasticity refers to a compensatory change in the organization of brain regions and can be harmful in cases such as tinnitus, which is called "plasticity disorder"[20]. In most cases of tinnitus, a loss of auditory input resulted in re-direction of information to areas not normally involved in sound processing [21]. The neural plasticity in tinnitus has been reported in the dorsal cochlear nucleus (DCN), inferior colliculus, hippocampus of the limbic system, parafloccular lobe (PFL) of the cerebellum, and auditory cortex [22]. Previous work has shown significant evidence of the involvement of DCN, limbic system and PFL regions in tinnitus [23–25]. DCN plays an important role in the production of tinnitus, and its neuronal hyperactivity has been reported as the most consistent finding of tinnitus in animal models [26,27]. The occurrence of plastic changes in the synapse-rich layers of the hippocampus has been confirmed in chronic tinnitus [28,29]. It has also been suggested a feedback loop between PFL and auditory cortex, the disruption of which can affect the persistence of tinnitus [24].

Doublecortin (DCX) is a microtubule-associated protein that contributes to synaptic regeneration and is known as a marker of neural plasticity. In adults, DCX has been reported to be expressed in the DCN, PFL, subventricular zone and dentate gyrus (DG) of the hippocampus [30,31].

Photobiomodulation therapy (PBMT) uses the light of laser or light emitting diodes (LEDs) to repair the structural disorders and modulate the function of target tissue. Light is first absorbed by the enzyme cytochrome c oxidase (CCO) in the mitochondria which facilitates electron transfer from CCO to $O_2$ and converts anaerobic phosphorylation in damaged tissue to oxidative phosphorylation with more ATP production [32]. As a conclusion, signaling molecules including reactive oxygen species (ROS), cyclic adenosine monophosphate (cAMP), $Ca^{+2}$ and nitric oxide (NO) increase, leading to the activation of transcription factors and increased gene expression. These genes play a role in differentiation, proliferation and migration of cells, as well as the synthesis of proteins, anti-apoptotic enzymes, and antioxidants. Light absorption also dissociates NO from CCO, which in pathological conditions competes

with oxygen for binding to the binuclear center of CCO. Free NO is a potent vasodilator that increases blood supply and oxygenation of the brain tissue [33]. Transcranial PBMT has been found to stimulate Brain-derived neurotrophic factor (BDNF) production which is a key modulator of neural plasticity events through the regulation of inhibitory γ-amino butyric acid (GABA) circuits [34,35].

PBMT has been used in several studies in the treatment of tinnitus, the limitations of which were the use of subjective evaluation tools, the use of a wide range of laser parameters, and neglecting the treatment of central compensatory changes [36–52]. There are two methods to confirm the occurrence of tinnitus in animal models. The operant method that requires the training and careful experimental control of the subjects and the reflexive method such as gap-prepulse inhibition acoustic startle (GPIAS) test which is based on the evaluation of a reflex response. This method is much faster, allows to separate tinnitus-positive from tinnitus-negative animals and can also be used by scientists with little experience in animal behavior [53]. At the moment, despite some limitations, several studies have used the GPIAS test and have led to significant findings in tinnitus research [54–56].

The ABR test assesses the neural response of the auditory pathway from the distal portion of eighth nerve to the inferior colliculus in the midbrain [57]. In the present study, this test was used to identify changes in the electrophysiological responses of the auditory pathway caused by tinnitus.

The aim of this study was to investigate the effects of PBMT on sodium salicylate (SS)-induced tinnitus using three objective assessment tests focusing on the changes of harmful neural plasticity in the three regions of DCN, DG and PFL.

## Material and methods

This study was conducted in accordance with the National Institutes of Health Guide for the Care and Use of Laboratory Animals and the ARRIVE guidelines and was approved by the Experimental Research Committee of the ENT and Head and Neck Research Center of The Five Senses Health Institute of IUMS, with registration number of 98-2-22-15523. Ethical Committee of Experimental Sciences of IUMS has confirmed the project with the ethical code "IR.IUMS.REC.1398.578".

A total of 21 rats were divided into 3 groups of control, tinnitus and PBMT, 7 rats in each group. Rats were raised under standard conditions and housed three rats per cage, with free access to food and water. The ambient temperature was 20±2˚C in a 12-hour light/dark cycle. Animal cages were cleaned at least twice a week. Before performing the behavioral and electrophysiological tests, the cages of the animals were placed in the test area for 30 minutes to adapt to the surrounding environment.

### Study design

GPIAS test was performed on the first, seventh and fifteenth days and ABR test was performed on the first and fifteenth days for all groups. In the first session, after the GPIAS and ABR tests, normal saline (NS) was injected to the control group, and SS was injected into the tinnitus and PBMT groups for the seven consecutive days [58]. In the 7th session, to confirm the occurrence of tinnitus in the tinnitus and PBMT groups and the absence of tinnitus in the control group, as well as to check the hearing status of the animals, GPIAS test was performed again for all groups. From the 7th session, PBMT was started once a day for 8 consecutive sessions for the PBMT group. No placebo treatment was done for the control group. SS injection in the tinnitus and PBMT groups, as well as NS injection in the control group, continued until the 14th day of the procedure. In the 15th day after performing the GPIAS and ABR tests, the rats

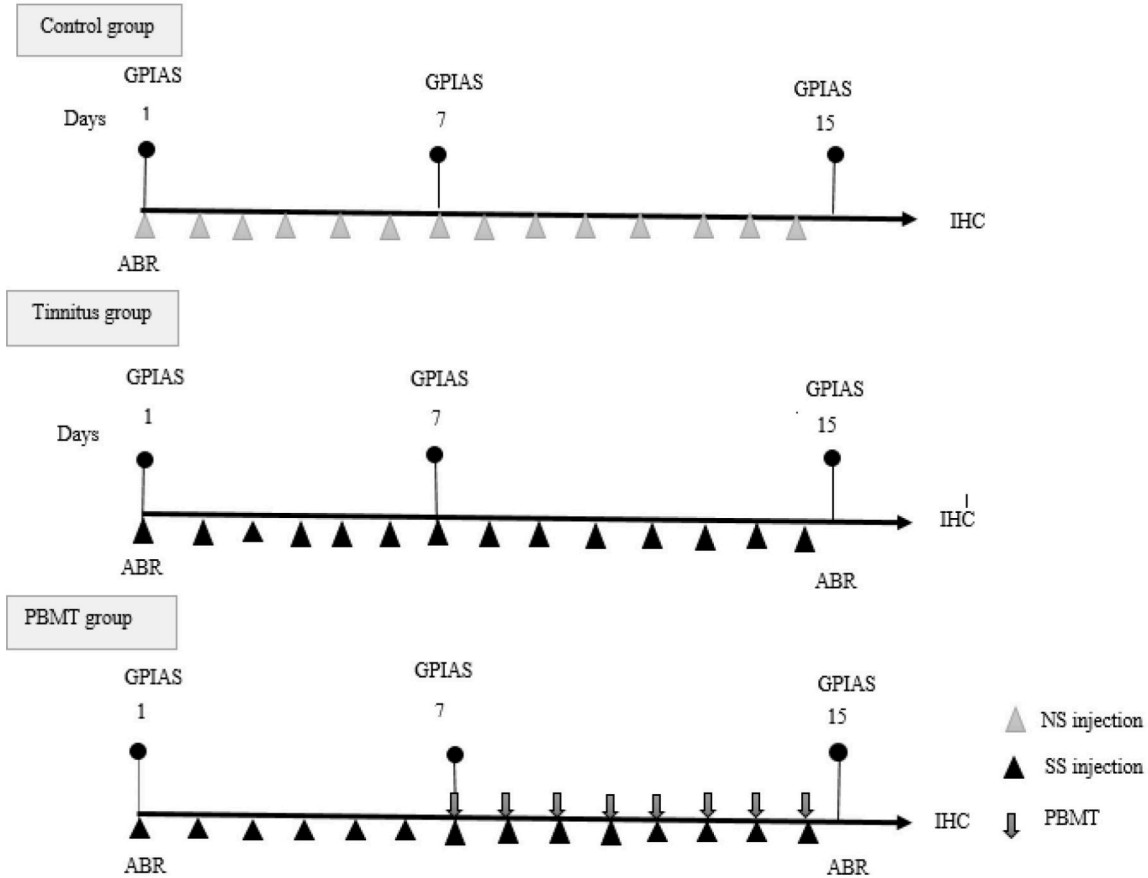

**Fig 1. Diagram of methods.** Abbreviations: GPIAS (Gap Prepulse Inhibition of Acoustic Startle), ABR (auditory brainstem response), SS (sodium salicylate), Cont. (control), PBMT (photobiomodulation therapy), IHC (immunohistochemistry).

were decapitated and their brains were fixed in 10% formalin and sent to the immunohistochemistry (IHC) laboratory to determine the expression of DCX in DCN, DG of hippocampus and PFL of cerebellum (Fig 1).

## Tinnitus induction method

Induction of tinnitus in tinnitus and PBMT groups was performed by intraperitoneal injection of 400 mg/kg sodium salicylate (Sigma-Aldrich, Shanghai, China) diluted in 5 cc/kg normal saline once a day for 7 consecutive days [58]. On the seventh day of injection, the values of GIN and PPI were measured using the GPIAS test. To maintain the tinnitus model, SS injection was continued until the 14th day of experiment.

## Anesthesia

The ethical review checklist for research projects involving laboratory animals based on the ethics committee of IUMS was followed in this study. The animals were anesthetized by intraperitoneal injection of ketamine 50 mg/kg and xylazine 10 mg/kg before the ABR test and the PBMT sessions. The state of anesthesia was assessed by gently pinching the toe of the animal every 10 minutes. If the rat does not respond to the toe pinch, it is considered to be unconscious. Some studied rats needed an additional anesthesia dose, which was injected along with

monitoring the animal's respiratory status. The rat is closely monitored during anesthesia to ensure that it remains adequately sedated throughout the procedure.

## Euthanasia of rats

Inhalation of carbon dioxide ($CO_2$) was used as a euthanasia technique in this study. Carbon dioxide is commonly used to euthanize rats by inducing hypoxia. Rats are placed in a chamber filled with $CO_2$, which displaces oxygen and leads to unconsciousness and death. The carbon dioxide gas flow rate was adjusted to 20% of the chamber volume per minute on the $CO_2$ gas flowmeter. Therefore, the flow rate of the flowmeter was set at 2 liters per minute (acceptable range: between 1 and 3 liters per minute). Usually, follow by 3 to 4 minutes, the animal loses consciousness. After confirming all the criteria for death, including arresting the heartbeat, stop of breathing movements, lack of reflexes, changing the color of the animal's eyes, coldness of the animal's body, and complete contraction of the animal's body muscles, the animal was decapitated.

## PBMT device and protocol

Laser device "MDL-lll-808" (CNI Co., China) producing infra-red wavelength of 808 nanometer (nm) was used as the light source for PBMT. The laser device has been coupled with a 200 micrometer (μm) fiber shielded in a metal tube. Vega Ophir laser power meter was used for checking the exact output power at the end of the fiber optic.

After the confirmation of tinnitus by GPIAS test in the seventh day of SS injection period, the treatment was performed for 8 consequent sessions. The wavelength of 808 nm was chosen due to its high penetration depth [59,60]. The power density of 165 mW/cm² as well as the number and duration of treatment sessions were chosen based on an animal study that used PBMT in SS-induced tinnitus in rats [61]. Due to the important role of power density in the safety and effectiveness of the treatment, the studies of using PBMT in acoustic trauma and hearing loss in animal models were also investigated in which the power density was in the range of 110 to 165 mW/cm² [62–64]. Using a power meter, the current (ampere) needed to produce a power density of 165 mW/cm² was determined. The power was measured at the optical fiber outlet, so the calculated energy density corresponds to the outlet of fiber. Central and peripheral areas involved in tinnitus were irradiated simultaneously, and to treat central compensatory changes, radiation to the whole brain was required. Laser irradiation can be done in contact or non-contact mode. The non-contact mode of radiation increases the waste of energy but reduces the possibility of tissue damage. The very small diameter of the optical fiber (200 μm) causes a lot of energy at the radiation site, so to avoid tissue damage and also to cover the entire brain area, the optical fiber was fixed at a distance of 8 cm from the animal's head. At this distance, the laser beam creates a circle with a diameter of 2 cm, which can cover the entire surface of the animal's brain. The final protocol included laser irradiation from the top to the brain and from the lateral side to the right and left ears for 10 minutes in each location "Table 1" (Fig 2).

## ABR test

In the ABR test, amplitude of waveforms, brainstem transmission time (BTT), SP/ AP ratio and threshold were the measured parameters. Amplitude refers to the distance between the peak and trough of each ABR wave in microvolt (μV), BTT refers to the time between the peak of wave I and the trough of wave V, which indicated the progress of nerve stimulation from the terminal part of the auditory nerve to the inferior colliculus of the brainstem. SP is the peak generated by the cochlear hair cells and is defined as the difference between the baseline

**Table 1. PBMT parameters.**

| | |
|---|---|
| **Wavelength** | 808 nm |
| **Mode of irradiation** | Continuous wave |
| **Power density** | 165 mW/cm$^2$ |
| **Energy density** | 99 J/cm$^2$ |
| **Total energy per session** | 297 J/cm$^2$ |
| **Number of sessions** | 8 |
| **Diameter of optical fiber** | 200 μm |
| **Duration of radiation per session** | 30 min |
| **Duration of radiation in each location** | 10 min |

Abbreviations: nm (nanometer), J (joules), mW (milliwatts), cm (centimeter), μm (micrometer), min (minutes).

and the last inflection point in the rising phase of wave I. AP is the action potential of the cochlear neurons, represented by the amplitude of wave I. A change in the SP/AP ratio is defined as one of the signs of hidden hearing loss [5]. Threshold is the lowest ABR sound pressure level (SPL) at which "uniform" peaks and troughs were observed.

The ABR test was performed using the "Audiology Lab" system (Otoconsult, Frankfurt am Main, Germany) and the waveforms were analyzed offline by MATLAB software to extract auditory evoked potentials. Click sound stimuli were presented by a calibrated speaker (DT48, Beyer Dynamic, Heilbronn, Germany) through a plastic cone placed in the rat's external auditory canal. The reference electrode was at the vertex, the active electrode was at the mastoid of the left ear, and the ground electrode was at the right ear. The sampling rate was set at 60 kHz,

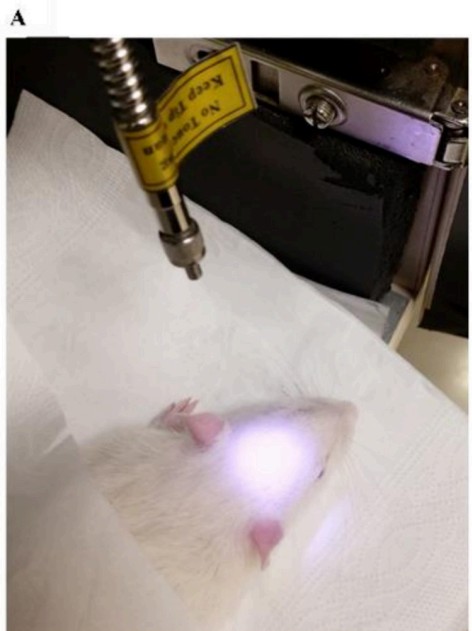 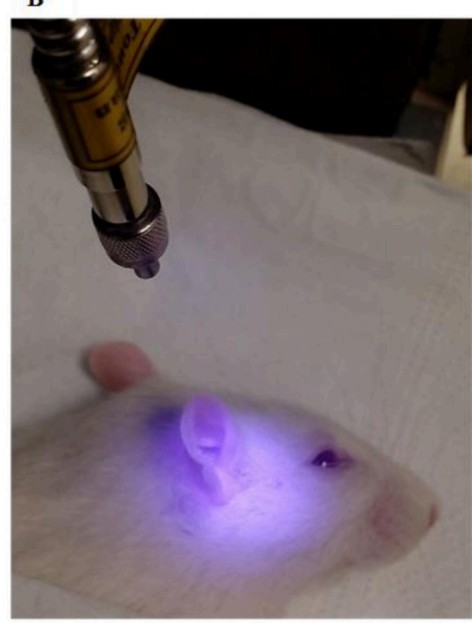

**Fig 2. Laser irradiation in the PBMT group.** The distance between the tip of the optical fiber and the animal's head was fixed at 8 cm by the mechanical holder, at which the diameter of the laser beam becomes 2 cm, which covers the entire surface of the animal's brain. (a) laser radiation to the skull from the upper side (b) laser radiation to the left and right hearing area from the lateral side.

and evoked potentials were filtered from 0.3 to 3.0 kHz. During the measurement, the sound pressure was decreased from 90 to 10 decibel (dB) in steps of 10 dB to determine the threshold.

### Behavioral test

The GPIAS behavioral test measured gap-in-noise (GIN) and prepulse inhibition (PPI) values using the SR-LAB-Startle Test Response System. The occurrence of tinnitus was assessed by measuring the GIN value, which measured the startle reflex in response to a sudden stimulus (pulse). The experiment consisted of 12 no-gap trials and 12 gap trials in which a silent gap was embedded in the soft background noise before the pulse. A normal animal can detect the gap in gap trials and its startle reflex is inhibited, but in tinnitus, the gap is filled with the buzzing sound and the startle reflex is not reduced. The GIN ratio indicates the inhibition percentage of the startle reflex and its significant reduction compared to before the injection of SS should be confirmed the occurrence of tinnitus [65]. The hearing status of the animals should be evaluated by PPI test because the GIN value will not be valid if the tinnitus-inducing drug has caused hearing loss in the animal. In the PPI test, a weak stimulus (prepulse) was played before the main pulse, which resulted in a reduction of the startle reflex in normal-hearing animals [65].

### IHC test

Whole brains were fixed in 10% formalin and brain tissues were embedded in paraffin. The samples were sectioned to a thickness of 5 μm and subjected to IHC staining. In short, following the tissue process (Churg 1983) Xylene and alcohol were used to deparaffinize and rehydrate the slides, then Slides were subjected for antigen retrieval by citrate buffer (PH = 6) and followed by incubation in 3% hydrogen peroxide for 10 minutes to block endogenous peroxidase activity. Then the sections were probed in Doublecortin (ab167400) (1:500) as a primary antibody which was diluted in 1% (w/v) skim milk in TBS-T [0/05 (v/v) Tween-20 in Tris-buffered saline] overnight at 4˚C. After washing the sections three times for 5 min in washing solution (1×TBS plus 0.03% Triton X-100), the slides were incubated in secondary antibody (Goat Biotinylated Polyvalent) (ab64261) for 10 min at RT. The sections were then washed three times for 1 min and incubated for 10 min in a streptavidin-conjugated horseradish peroxidase (ab64261). Another three washes of 1 min were performed and the sections were treated with a DAB substrate, until the tissue was colored (1–10 min). Then the sections were washed three times for 1 min and counterstained with hematoxylin for 2 min in RT, then underwent dehydration in graded ethanol, were cleared in xylene, and cover slipped with Permount mounting media.

### Statistical analysis

The SPSS version 16 (IBM-SPSS, Inc., Chicago, IL, USA) was used for statistical analysis. To evaluate the data distribution, the Kolmogorov-Smirnov test was performed, which confirmed the normal distribution. Descriptive analysis, Analysis of Variance (ANOVA) and Scheffe tests were performed for between group comparison and Prism version 8 was used to draw the graphs. P-value $\leq$ 0.05 was considered significant.

**Semi-quantitative analysis.** Brain sections were evaluated and scored by evaluating the expression level of DCX antibody (Anti-Doublecortin antibody [EPR10935(B)]) in the studied groups. DCX expression was qualified by microscopic (Germany-AXIOM, BM-600 LED EPI FLURESCENT) equipped with a digital camera (Mshot, Chinese) at 40× and 200× magnification. The percentage of area occupied by DCX protein was determined in 3 experimental fields. Semi-quantitative analysis was done by calculating the average pixel intensity of 10

different and randomly selected fields per slide using Image J Fiji software, as described by Crowe and Yue [66]. One-way ANOVA and Tukey's test were used for statistical analysis of data.

## Results

### GPIAS test

Animals from the tinnitus group showed the reduction of the mean GIN value, compared to the control group. However, animals from the PBMT group showed an increase of the mean GIN value, compared to the tinnitus group. Therefore, there was a significant difference at the P$\leq$ 0.05 level between the groups in terms of the GIN value (F (2,18) = 7.637, P = .004). For PPI value, difference the between groups was not significant (F (2,18) = 1.468, P = 0.257) (Fig 3).

### Electrophysiological (ABR) test

Recorded ABR waveforms of the control, tinnitus and PBMT groups are shown in (Fig 4).

**Statistical comparison of ABR characteristics.** As it has been shown in Fig 4, wave II of ABR was the most prominent waveform in all groups. Since the highest mean amplitude of this waveforms was created at 90 dB SPL, the statistical comparisons in terms of the amplitude of waveforms, SP/AP ratio, threshold and BTT have been made for 90 dB SPL.

**ABR threshold and BTT.** Animals from the tinnitus group showed the elevation of the mean BTT, compared to the control group. However, animals from the PBMT group showed a decrease of the mean BTT compared to the tinnitus group. Therefore, there was a significant difference at the P$\leq$ 0.05 level between the groups [F (2, 18) = 11.3423, P = .001].

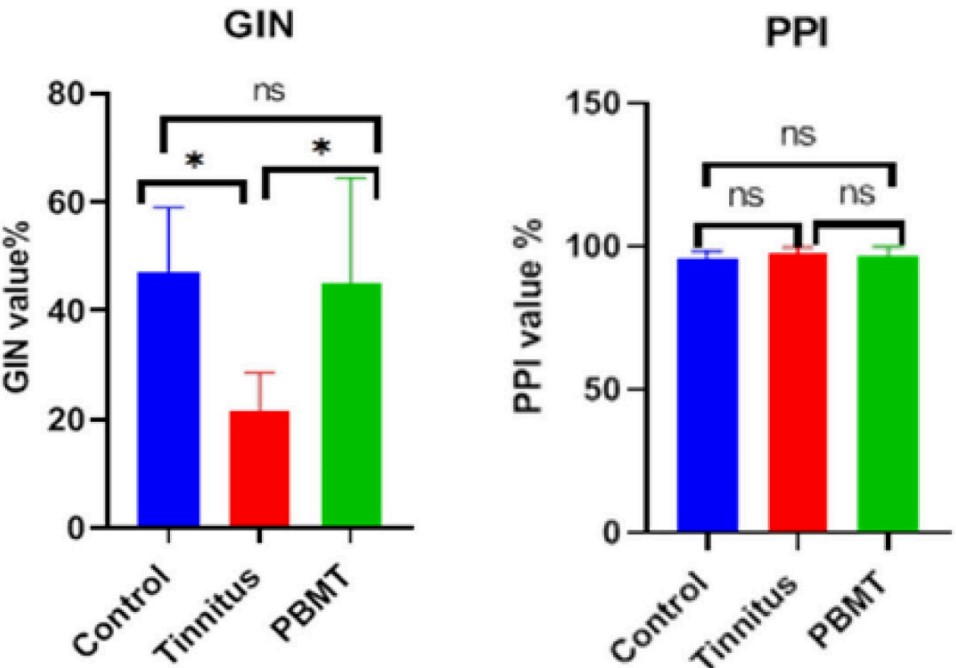

**Fig 3. Between-groups comparison of GIN and PPI values in control, tinnitus and PBMT groups.** Abbreviations: GIN (gap-in-noise), PPI (pre-pulse inhibition), PBMT (photobiomodulation therapy).

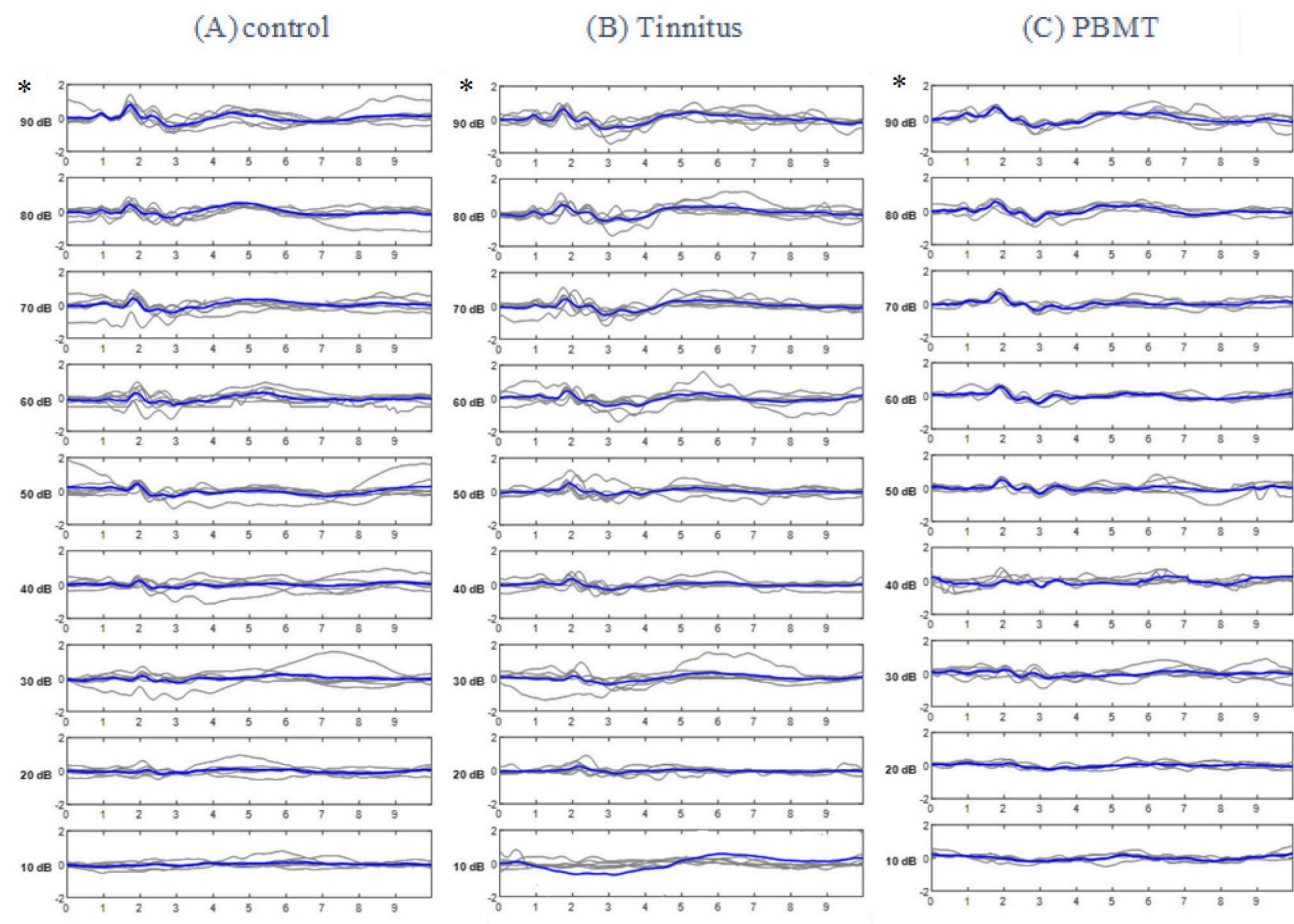

**Fig 4.** ABR waveforms in (A) Control group (B) Tinnitus group C) PBMT group. * The highest mean amplitude was produced at 90 dB SPL and in wave II of ABR. In each row, the bold line represents the grand average of the waveforms.

Animals from the tinnitus group showed an increase of the mean threshold, compared to the control group. However, animals from PBMT group showed a decrease of the threshold, compared to the tinnitus group. Therefore, there was a significant difference at the P≤ 0.05 level between 3 groups [F (2,18) = 8.712, P = .002] "Table 2" (Fig 5).

**Amplitude of waveforms, SP/AP ratio.** There was not a significant difference in wave I amplitude at the P ≤ 0.05 between the control, tinnitus and PBMT groups [F (2, 18) = 0.165, P = 0.849)]. There was not a significant difference in wave II amplitude between the groups [F (2, 18) = 0.172, P = 0.843)]. There was not a significant difference in wave III amplitude between the groups [F (2, 18) = 0.176, P = 0.838)]. There was not a significant difference in

**Table 2. Mean±SD of BTT and threshold in control, tinnitus and PBMT groups.**

| Groups (M±SD) | BTT (ms) | Threshold (dB) |
|---|---|---|
| Control | 4.11±0.10 | 22.8±7.5 |
| Tinnitus | 5.18±0.61 | 35.7±11.3 |
| PBMT | 4.59±0.38 | 15.7±7.8 |

Abbreviations: M (mean), SD (standard deviation), ms (milli seconds), dB (decibel).

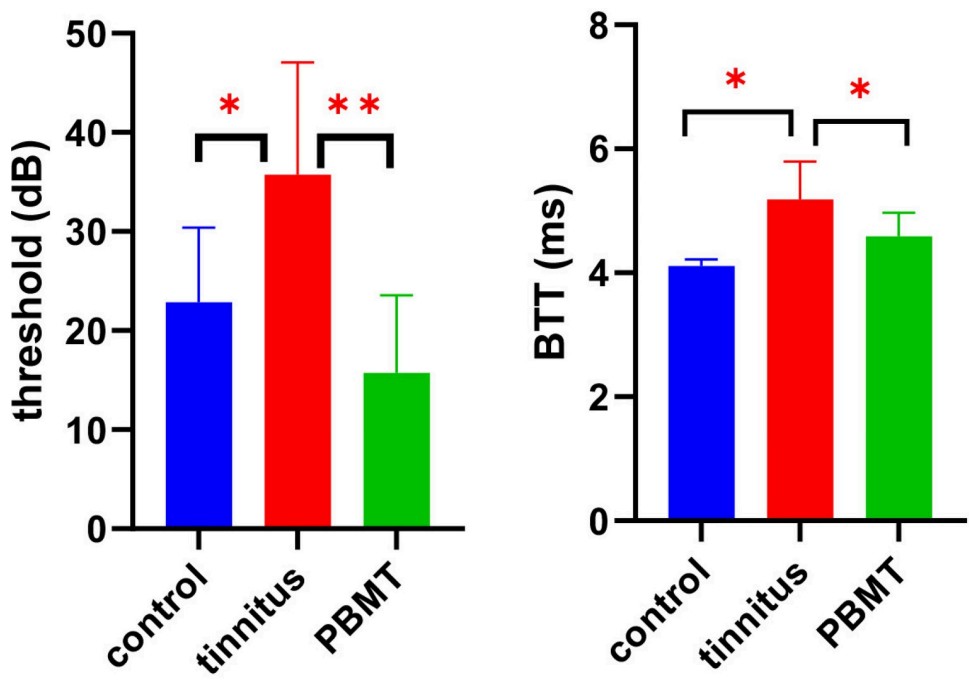

**Fig 5. Between groups comparison of BTT and threshold in control, tinnitus and PBMT groups.** Abbreviations BTT (brainstem transmission time), PBMT (photobiomodulation therapy), dB (decibel), ms (millisecond).

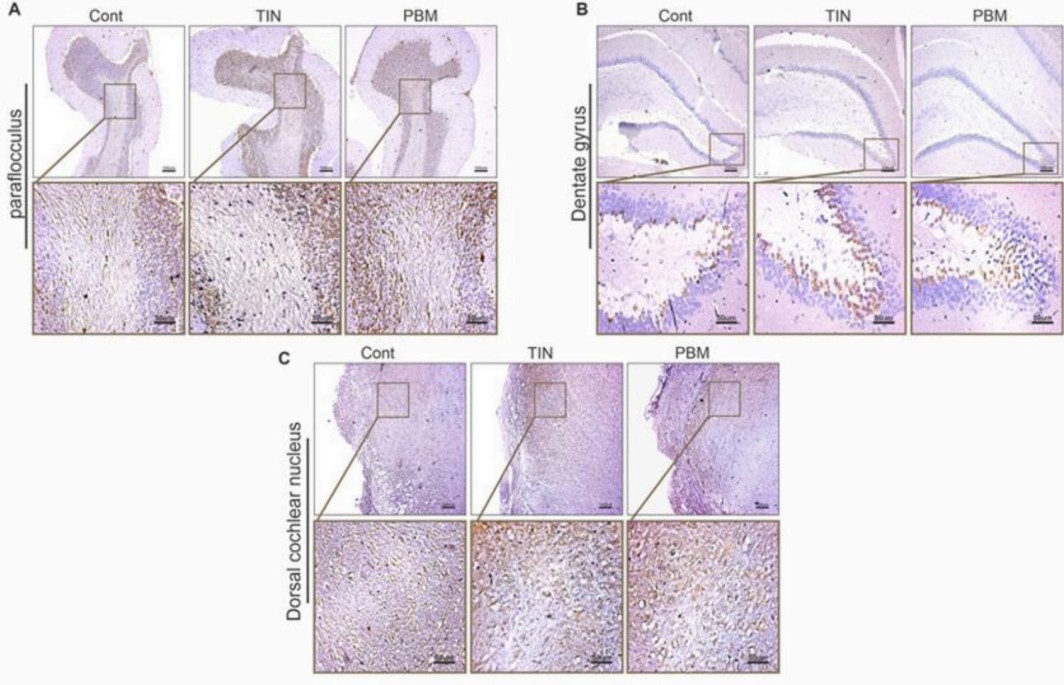

**Fig 6. Doublecortin (DCX) expression in 3 brain regions (coronal sections) in the control (Cont), Tinnitus (TIN) and PBMT (PBMT) groups.** DCX expression indicated as brown infiltration. (A) DCX expression in the paraflocculus (B) DCX expression in the dentate gyrus (C) DCX expression in the dorsal cochlear.

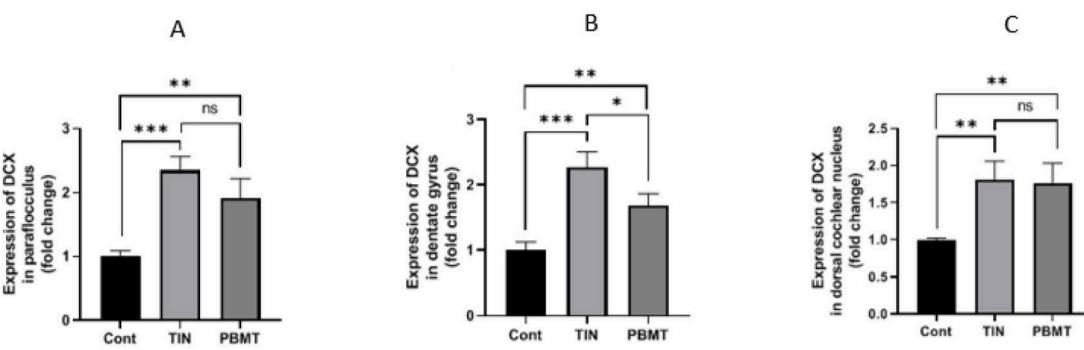

**Fig 7. Bar graphs of DCX expression evaluated in three brain regions in the control (Cont), Tinnitus (TIN) and PBMT (PBMT) groups.** (A) DCX expression in Paraflocculus (B) DCX expression in dentate gyrus (C) DCX expression in dorsal cochlear nucleus.

wave IV amplitude between the groups [$F_{(2, 18)} = 0.161$, $P = 0.815$]. There was not a significant difference in wave V amplitude between the groups [$F_{(2, 18)} = 0.183$, $P = 0.829$]. There was not a significant difference in SP/AS ratio between the groups [$F_{(2, 18)} = 0.084$, $P = 0.918$].

### IHC test

In the DG, the expression of DCX in the tinnitus group was significantly higher than the control group ($P = 0.0004$). In the PBMT group, the expression of DCX was significantly higher than the control group ($P = 0.009$), but it was significantly lower than the tinnitus group ($P = 0.01$). In the DCN, the expression of DCX in the tinnitus group was significantly higher than the control group ($P = 0.006$). The expression of DCX in the PBMT group, was significantly higher than the control group ($P = 0.009$). The expression of DCX was not significantly different between tinnitus and PBMT group ($P > 0.05$).

In the PFL, the expression of DCX in the tinnitus group was significantly higher than the control group ($P = 0.0008$) and in the PBMT group it was significantly higher than the control group ($P = 0.005$). The difference in DCX expression between tinnitus and PBMT groups was not significant ($P > 0.05$) (Figs 6 and 7).

### Discussion

PBMT is a safe, accessible, and easy-to-use therapeutic tool and this study was an attempt to confirm its effectiveness in the treatment of tinnitus as a common and complex condition that has no cure. To overcome the limitations of previous studies, necessary objective evidence was provided by using three objective evaluation tests. The most interesting findings were the consistency of the results of the evaluation tests as well as the reduction of the harmful plasticity caused by tinnitus in the treatment (PBMT) group.

In animal models, tinnitus is usually induced by exposure to noise or injection of SS. In noise exposure, the different protocols may lead to different neurophysiological changes and the use of anesthesia during noise exposing may protect against tinnitus. SS is widely used to induce tinnitus in animal models and has been found to decrease outer hair cell electrical conductance as well as GABA transmission [67]. It was found that in rats, induction of tinnitus by SS is highly reliable whereas noise trauma has a much more variable success [68]. The first finding was the reduction of the mean GIN value in the tinnitus group compared to the control group, which indicated the creation of an additional perceived sound like tinnitus in animals

[65]. Importantly in the PBMT group, the mean GIN was significantly increased in compare to the tinnitus group (Fig 3). The GIN value calculated the percentage of inhibition of the startle reflex, and its increase in the PBMT group indicated that the animal was able to detect the gap in the background noise, possibly due to the interruption or reduction of the buzzing sound. In this study, the significant increase in GIN value in the PBMT group compared to the tinnitus group was the first indication of the effectiveness of PBMT in tinnitus.

The PPI value remained within the normal range in all three groups, indicating that the hearing level of the animals was not affected by SS injection (Fig 3). Normal PPI values at the tinnitus-inducing dose of SS have also been reported in other studies [53,54,58,69].

As mentioned, hidden hearing loss is a type of hearing loss that cannot be detected by routine audiometric tests and is caused by a cochlear synaptopathy. It has been reported that exposure to SS can cause loss of cochlear ribbon synapses connecting cochlear inner hair cells and spiral ganglion cells [70]. Considering the possible effects of SS on cochlear synapses as well as the normal hearing level, the animals were examined for the possibility of hidden hearing loss. In previous studies, reduction in peak I amplitude and change in SP/AP ratio have been reported as signs of hidden hearing loss in ABR testing in animal models [5,7,8]. In our experiment, there was no significant difference in wave I amplitude and SP/AP ratio between the groups, so the occurrence of hidden hearing loss was not demonstrated. It should be considered that in previous studies, the ototoxic drugs that were used to induce hidden hearing loss in animal models were all from the aminoglycoside group and SS was not used [5,7,8].

Another finding of ABR testing was the increase of the mean values of BTT and threshold in the tinnitus group and more importantly, the return of both parameters to the normal level in the PBMT group (Fig 5). An increase in ABR threshold with a tinnitus-inducing dose of SS has also been reported in other animal studies [71,72]. A possible explanation for the increased threshold could be the mild mechanical disturbance caused by the SS injection in the cochlear outer hair cells. It has been reported that the compression exerted by the outer hair cells on the basement membrane increases the dynamic range of the cochlea up to 120 dB, and the dysfunction of these cells due to SS injection may reduce the distance between the minimum audible sound level (threshold) and the maximum tolerable sound level [73].

The prolongation of BTT in the tinnitus group may be explained by the effect of SS on sodium channels. It has been reported that a tinnitus-inducing dose of SS may cause a concentration-dependent block of the voltage-gated sodium channels. The activity of these channels is essential for the conduction and excitability of all neurons, as they mediate the fast-rising phase and the early-falling phase component of action potentials in many excitable cells [74]. The effect of SS on the voltage-gated sodium channels may be one of the reasons of BTT prolongation which needs further studies to be confirmed.

As mentioned, both BTT and threshold of ABR returned to normal level in PBMT group. It has been found that PBMT has the ability to prevent the loss of cochlear hair cells as well as restore them in acoustic trauma and the use of ototoxic drugs [62,63,75]. Furthermore, PBMT can be a potential therapeutic tool for disorders involving Na, K-ATPase pumping [76]. In terms of the neurological function, it has been confirmed that PBMT has the ability to accelerate nerve regeneration, improve electrophysiological function and functionality rate, as well as the release of growth factors and increase of vascular network and collagen [77]. Of course, more studies are needed to determine the causes and mechanism of re-normalization of BTT and threshold after PBMT.

An important finding was a significant increase in DCX expression in DCN, PFL, and DG in the tinnitus group (Figs 6 and 7). This finding is consistent with the decrease of mean GIN value which was the finding of the GPIAS test in confirming tinnitus. Although many animal studies have used the GPIAS test to confirm the occurrence of tinnitus, its validity has been

challenged in some studies [53,78,79]. One of the drawbacks attributed to GPIAS is that no inhibition should occur at all if the tinnitus indeed fills the gap, whereas inhibition of the startle reflex has been observed in some cases [53,79]. In addition, some studies have suggested that inhibition reduction in tinnitus is likely to occur only when the background noise is qualitatively similar to the tinnitus [78]. The compatibility of the results of GPIAS and IHC tests in the current study may support the validity of GPIAS test in tinnitus evaluation.

Pathophysiological changes in tinnitus have been confirmed to be associated with harmful neural plasticity. Evidence suggests that immediate and long-term activation of non-classical auditory structures (such as the amygdala, hippocampus, and cingulate cortex) may play an important role in the initiation, development, and persistence of tinnitus [80]. The increase in the expression of DCX as a marker of neural plasticity in the three mentioned regions was an important finding that indicated the occurrence of plastic changes in tinnitus and also the role of these regions in the development of tinnitus (Figs 6 and 7). Increased expression of DCX in the DCN and PFL immediately after noise exposure has been previously reported [25]. In another study, the expression of DCX was evaluated one month after exposure to noise and it was reported to decrease in DG, increase in PFL, and remain unchanged in DCN [23]. The reduced expression of DCX in the DG and its unchanged expression in the DCN one month after noise exposure could indicate an important role of assessment time and possibly the brain's compensatory ability to re-normalize plastic changes over time, which is still needed further researches to be confirmed.

The most important finding of this study was the significant reduction of DCX expression in DG in the PBMT group (Figs 6 and 7). DCX expression was also reduced in the two other regions but the reduction was not become significant in statistical analysis. Reportedly, PBMT has been found to stimulate BDNF production which is a critical regulator of synaptic plasticity that promotes synaptic transmission and synaptogenesis [35]. PBMT has also been shown to improve synaptic plasticity in Alzheimer's disease [81]. The ability of PBMT to stimulate BDNF may explain its effects on reducing harmful neuroplasticity in tinnitus, however further research is needed to confirm this. According to the gatekeeper theory of tinnitus, the limbic system has the potential to prevent the transmission of tinnitus signals to the auditory cortex. However, if the limbic structures fail to block the hyperactive signals of the classical auditory pathways, this "filtering failure" will lead to the perception of chronic tinnitus [19]. On the other hand, animal models of tinnitus induced by SS or noise have shown an early onset of limbic system plasticity within hours, minutes or days after the initial insult [19,80]. The mentioned evidence may help to explain the significant decrease of neuroplasticity in DG immediately after the completion of treatment sessions.

## Limitations of the study

To comply with ethical principles, the minimum accepted number of animals was used, and the small sample size was one of the limitations of this study.

To optimize the PBMT protocol, we had to use two animals each time to determine the effectiveness of the protocol who were anesthetized and injected by SS so could not be used for other studies.

To cover the entire surface of the brain, the laser device was fixed at a distance of 8 cm from the animal's head, which increases energy dissipation. Laser helmet is suggested for future studies.

In this study, IHC test was performed only for two brain regions and DCN. For future studies, the IHC test can be performed in the peripheral auditory components such as cochlea to evaluate the effects of PBMT on hair cells.

## Conclusion

In this controlled study, PBMT was used in the simultaneous treatment of peripheral auditory defects and central compensatory changes in tinnitus, and the results were evaluated with behavioral, electrophysiological and IHC tests. In the tinnitus group, a significant decrease in the value of GIN in the behavioral test, a significant increase of BTT and threshold in ABR, and a significant increase of harmful plasticity in the IHC test were observed. In the PBMT group, there was a significant increase in GIN value, a significant decrease in BTT and threshold, as well as a significant decrease in harmful plasticity in the DG. According to our findings, PBMT has the potential to be used as a therapeutic tool in SS-induced tinnitus. The effect of PBMT in reducing harmful plasticity and normalizing the value of GIN, BTT and threshold emphasizes the necessity of simultaneous treatment of central and peripheral defects in tinnitus, as well as delivering sufficient energy to auditory and non-auditory areas of the brain. Our findings open a new horizon in the use of PBMT in the treatment of tinnitus, and further studies in this field can be beneficial.

## Acknowledgments

The authors gratefully acknowledge the Center of Experimental and Comparative Medical Studies colleagues of Iran Universit of Medical Sciences (IUMS), Tehran, Iran, for providing the laboratory animals and permission to use the SR-LAB-Startle test response system to perform the present study.

## Author Contributions

**Conceptualization:** Katayoon Montazeri, Mohammad Farhadi, Reza Fekrazad, Ali Shahbazi, Saeid Mahmoudian.

**Data curation:** Katayoon Montazeri, Abbas Majdabadi, Zainab Akbarnejad, Ali Shahbazi.

**Formal analysis:** Abbas Majdabadi, Zainab Akbarnejad, Saeid Mahmoudian.

**Investigation:** Katayoon Montazeri, Zainab Akbarnejad.

**Methodology:** Katayoon Montazeri, Abbas Majdabadi, Zainab Akbarnejad, Reza Fekrazad, Ali Shahbazi, Saeid Mahmoudian.

**Project administration:** Mohammad Farhadi, Saeid Mahmoudian.

**Resources:** Saeid Mahmoudian.

**Software:** Abbas Majdabadi.

**Supervision:** Mohammad Farhadi, Reza Fekrazad, Saeid Mahmoudian.

**Validation:** Mohammad Farhadi, Abbas Majdabadi, Zainab Akbarnejad, Reza Fekrazad.

**Visualization:** Ali Shahbazi.

**Writing – original draft:** Katayoon Montazeri, Saeid Mahmoudian.

**Writing – review & editing:** Katayoon Montazeri, Saeid Mahmoudian.

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
