## [Decision Letter · Decision Letter 0]

25 Jan 2024

PONE-D-23-42866Title: Photobiomodulation therapy in improvement of harmful neural plasticity in sodium salicylate-induced tinnitusPLOS ONE

Dear Dr. Mahmoudian,

Thank you for submitting your manuscript to PLOS ONE. After careful consideration, we feel that it has merit but does not fully meet PLOS ONE’s publication criteria as it currently stands. Therefore, we invite you to submit a revised version of the manuscript that addresses the points raised during the review process.

Please pay particular attention to the many points raised by Reviewer 2 as well as Reviewer 1

We look forward to receiving your revised manuscript.

Kind regards,

Michael R Hamblin

Academic Editor

PLOS ONE

“Special thanks to Iran National Science Foundation (INSF) (project number of 91058320, grant code no. insf-98020383-1400/03/23). With grateful of Iran University of Medical Sciences (IUMS), research & Technology deputy. Thankfulness to the colleagues of the "Center of Experimental and Comparative of Medical Studies of IUMS".”

5. In the online submission form you indicate that your data is not available for proprietary reasons and have provided a contact point for accessing this data. Please note that your current contact point is a co-author on this manuscript. According to our Data Policy, the contact point must not be an author on the manuscript and must be an institutional contact, ideally not an individual. Please revise your data statement to a non-author institutional point of contact, such as a data access or ethics committee, and send this to us via return email. Please also include contact information for the third party organization, and please include the full citation of where the data can be found.

Reviewers' comments:

Reviewer's Responses to Questions

**Comments to the Author**

1. Is the manuscript technically sound, and do the data support the conclusions?

Reviewer #1: Yes

Reviewer #2: Partly

2. Has the statistical analysis been performed appropriately and rigorously? 

Reviewer #1: I Don't Know

Reviewer #2: N/A

3. Have the authors made all data underlying the findings in their manuscript fully available?

Reviewer #1: Yes

Reviewer #2: Yes

4. Is the manuscript presented in an intelligible fashion and written in standard English?

Reviewer #1: Yes

Reviewer #2: Yes

5. Review Comments to the Author

Reviewer #1: I enjoyed reading this paper. Well done.

Please ensure that the manuscript is thoroughly checked for English expression and grammar.

For example:

line 57 perception of sound THAT occurs

line 72 "needed for widespread use" needs to be explained more clearly

the paragraph from line 78 is very clumsy

line 99 "In scientific sources" is better expressed as "PREVIOUS WORK HAS SHOWN"

line 128 "In animal models, tinnitus IS USUALLY INDUCED by"

line 114 the sentence beginning with "Then, during ...." appears incomplete or clumsy

line 188 is "should be" the correct way to express this

line 358 replace "common, complex and cure-less symptom" with something like "common AND complex CONDITION THAT HAS NO CURE"

line 356 I am not sure that "The GIN value represented the percentage inhibition", rather that the GIN ratio was calculated by ...?

line 379 & 382 replace "in the review articles" with "in PREVIOUS STUDIES"

line 382 replace not confirmed with NOT DEMONSTRATED

line 441 replace "needed to be confirmed " with "NEEDED TO CONFIRM THIS"

In the methods section, generally the study design is followed by the treatments (anaesthesia etc, PBMt) followed by outcome measures, and lastly followed by statistics and other analyses.

The table of PBMt parameters lists energy density as 99J/cm2. This is at the optical fibre outlet? Was there any effort to measure the energy density that the animal actually received? This needs to be addressed

I think that the last paragraph of the Discussion beginning line on 449 would fit better as the concluding paragraph of the Conclusion

Reviewer #2: Introduction

The introduction is too long. The information on plasticity could be reduced. Also, information on the method for confirming the occurrence of tinnitus could be reported in the methodology, or in the discussion, to clarify the findings with the model used.

Methods

The anesthesia topic is before the photobiomodulation topic. Please provide the information in chronological order in your methodology. In addition, your methodology should end with the topic of statistical analysis.

Figure 1 does not have a good resolution. Please include a figure with a good resolution.

Did the animals in the control group receive the placebo treatment? If so, include this information in the methodology. How was this procedure carried out?

Results

Maintain in the graphs only the symbols that represent a significant difference. Remove the bars with the abbreviation “ns”. In addition, the resolution of the graphs needs to be improved.

I suggest that the Electrophysiological (ABR) test topic be before the GPIAS test. This seems to be more consistent with your experimental design.

Include the ANOVA value in the description of the results. If so, you could use a description that indicates the observed phenomenon more precisely. For example: Animals from tinnitus group showed the reduction of the mean GIN value, compared to the control group. However, Animals from photobiomodulation group showed an increase of the mean GIN value, compared to the control group.

When describing the results, clearly present the ANOVA values and the post hoc values

The figures need a better resolution

Discussion

Insert the topic limitations of the study at the end of the discussion.

6. PLOS authors have the option to publish the peer review history of their article (what does this mean?). If published, this will include your full peer review and any attached files.

Reviewer #1: No

Reviewer #2: No

---

## [Author Response · Author response to Decision Letter 0]

24 Feb 2024

Reviewer#1

- Line 57 perception of sound THAT occurs √ Done

- Line 72 "needed for widespread use" needs to be explained more clearly√ the paragraph was changed

- The paragraph from line 78 is very clumsy: √the paragraph was changed

- Line 99 "In scientific sources" is better expressed as "PREVIOUS WORK HAS SHOWN"√

- Line 128 "In animal models, tinnitus IS USUALLY INDUCED by"√

- Line 114 the sentence beginning with "Then, during ...." appears incomplete or clumsy √ the sentence was changed

- Line 188 is "should be" the correct way to express this√

- Line 358 replace "common, complex and cure-less symptom" with something like "common AND complex CONDITION THAT HAS NO CURE"√ Was replaced

- Line 356 I am not sure that "The GIN value represented the percentage inhibition", rather that the GIN ratio was calculated by ...? √ Done

- Line 379 & 382 replace "in the review articles" with "in PREVIOUS STUDIES"√ Done

- Line 382 replace not confirmed with NOT DEMONSTRATED√ Done

- Line 441 replace "needed to be confirmed " with "NEEDED TO CONFIRM THIS"√ Done

- In the methods section, generally the study design is followed by the treatments (anesthesia etc., PBMT) followed by outcome measures, and lastly followed by statistics and other analyses. √ relocation done

- The table of PBMT parameters lists energy density as 99J/cm2. This is at the optical fiber outlet? Was there any effort to measure the energy density that the animal actually received? This needs to be addressed√ it was addressed

- I think that the last paragraph of the Discussion beginning line on 449 would fit better as the concluding paragraph of the Conclusion√ Relocation done

Reviewer # 2

-The introduction is too long. The information on plasticity could be reduced. √ Done 

- Also, information on the method for confirming the occurrence of tinnitus could be reported in the methodology, or in the discussion, to clarify the findings with the model used. √ Relocation done

- The anesthesia topic is before the photobiomodulation topic. Please provide the information in chronological order in your methodology. In addition, your methodology should end with the topic of statistical analysis. √ Done

- Figure 1 does not have a good resolution. Please include a figure with a good resolution. √ Done

Did the animals in the control group receive the placebo treatment? If so, include this information in the methodology. How was this procedure carried out? √ It was explained that we did not have placebo group.

- Maintain in the graphs only the symbols that represent a significant difference. Remove the bars with the abbreviation “ns”. We did not just remove the ns symbols from the graph bars because the non-significance of the difference between the GIN value of the PBMT group compared to the control group indicates the response to treatment, which is very important in this study and we wanted to highlight it. In addition, the non-significance of the difference between PPI values among groups which indicates the normal hearing is also important in our study. However, please let us know if you insist that these marks be removed and we will respectfully remove all ns marks. 

- In addition, the resolution of the graphs needs to be improved. √ Done

- I suggest that the Electrophysiological (ABR) test topic be before the GPIAS test. This seems to be more consistent with your experimental design. √ Done

- Include the ANOVA value in the description of the results. If so, you could use a description that indicates the observed phenomenon more precisely. For example: Animals from tinnitus group showed the reduction of the mean GIN value, compared to the control group. However, Animals from photobiomodulation group showed an increase of the mean GIN value, compared to the control group. √ Done

When describing the results, clearly present the ANOVA values and the post hoc values√ Done

The figures need a better resolution√ Done

Insert the topic limitations of the study at the end of the discussion. √ Done

---

## [Decision Letter · Decision Letter 1]

13 Mar 2024

Title: Photobiomodulation therapy in improvement of harmful neural plasticity in sodium salicylate-induced tinnitus

PONE-D-23-42866R1

Dear Dr. Mahmoudian,

We’re pleased to inform you that your manuscript has been judged scientifically suitable for publication and will be formally accepted for publication once it meets all outstanding technical requirements.

Kind regards,

Michael R Hamblin

Academic Editor

PLOS ONE

Additional Editor Comments (optional):

Reviewers' comments:

Reviewer's Responses to Questions

**Comments to the Author**

1. If the authors have adequately addressed your comments raised in a previous round of review and you feel that this manuscript is now acceptable for publication, you may indicate that here to bypass the “Comments to the Author” section, enter your conflict of interest statement in the “Confidential to Editor” section, and submit your "Accept" recommendation.

Reviewer #1: All comments have been addressed

2. Is the manuscript technically sound, and do the data support the conclusions?

Reviewer #1: Yes

3. Has the statistical analysis been performed appropriately and rigorously? 

Reviewer #1: I Don't Know

4. Have the authors made all data underlying the findings in their manuscript fully available?

Reviewer #1: Yes

5. Is the manuscript presented in an intelligible fashion and written in standard English?

Reviewer #1: Yes

6. Review Comments to the Author

Reviewer #1: All comments addressed. Thank you for your attention to this.

The report is a valuable addition to the PBM field.

7. PLOS authors have the option to publish the peer review history of their article (what does this mean?). If published, this will include your full peer review and any attached files.

Reviewer #1: No

---

## [Editor Report · Acceptance letter]

3 Apr 2024

PONE-D-23-42866R1 

PLOS ONE

Dear Dr. Mahmoudian, 

I'm pleased to inform you that your manuscript has been deemed suitable for publication in PLOS ONE. Congratulations! Your manuscript is now being handed over to our production team.

Kind regards, 

on behalf of

Dr. Michael R Hamblin 

Academic Editor

PLOS ONE